# An Evaluation Study of a New Designed Oscillating Hydraulic Trainer of Neck

**DOI:** 10.3390/healthcare11101518

**Published:** 2023-05-22

**Authors:** Hongchun Yang, Yawei Lv, Sisi Chen, Baixi Xing, Jianfeng Wu

**Affiliations:** 1Design and Research Institute, Zhejiang University of Technology, Hangzhou 310023, China; yhc2016@zjut.edu.cn (H.Y.); xingbaixi@zjut.edu.cn (B.X.); 2School of Design and Architecture, Zhejiang University of Technology, Hangzhou 310023, China; 2112015031@zjut.edu.cn (Y.L.); 2112015079@zjut.edu.cn (S.C.)

**Keywords:** strength training, neck flexion and extension, sEMG, ergonomics evaluation

## Abstract

In view of the importance of neck strength training and the lack of adequate training equipment, this study designed a new oscillating hydraulic trainer (OHT) of neck based on oscillating hydraulic damper. We used surface electromyography (sEMG) and subjective ratings to evaluate the neck OHT and compared the results with a simple hat trainer (HATT) and traditional weight trainer (TWT) to verify the feasibility and validity of the OHT. Under similar exercise conditions, 12 subjects performed a set of neck flexion and extension exercise with these 3 trainers. The sEMG signals of targeted muscles were collected in real time, and subjects were asked to complete subjective evaluations of product usability after exercise. The results showed that the root mean square (RMS%) of sEMG indicated that the OHT could provide two-way resistance and train the flexors and extensors simultaneously. The overall degree of muscle activation with OHT was higher than that with the other two trainers in one movement cycle. In terms of resistance characteristics exhibited by the sEMG waveform, duration (D) with OHT was significantly longer than HATT and TWT when exercising at a high speed, while Peak Timing (PT) was later. The ratings of product usability and performing usability of OHT were remarkably higher than that of HATT and TWT. Based on the above results, the OHT was proved to be more suitable for strength training, such as neck muscles, which were getting more attention gradually, but lacked mature and special training equipment.

## 1. Introduction

The incidence and burden of neck injury has gradually attracted people’s attention. Individual and social costs of severe head and neck injury could be very high [1]. The incidence of neck disease is related to the type of work engaged; dentists, nurses, and office workers have a higher prevalence of neck injury [2,3]. Computer workers have the highest incidence of neck injury, higher than what is observed in the general population [4]. This may be due to high levels of sedentary behavior, prolonged static muscular contraction and cervical loading, extreme working postures, poor ergonomics, and repetitive tasks, which increase the risk for developing neck injury [5]. In a study of the two national health services (NHS) hospitals in the UK, 10.8% of all emergency department visits related to sports were head and neck injuries [6]. The risk of sustaining a sports-related head or neck injury is greater in sports that involve body contact, projectiles, additional equipment (such as a hockey stick), and high speeds [7]. In addition, neck disorders caused by flight have been afflicting pilots for a long time. In flight effects, such as acceleration, sedentary behavior, head-worn equipment, and seatback, all are risk factors for neck pain and cervical spondylosis in pilots [8,9].

It is generally believed that neck strengthening may result in fewer neck injuries [10,11]. In recent years, there has been mounting interest in neck strength training plans [12]. For example, a retrospective analysis of a squad of professional rugby union players found that when isometric neck strengthening was incorporated into the overall conditioning program, there was a reduction in the number of match-related cervical spine injuries compared with the previous season [13]. Different types of equipment were utilized in the training programs. High school football players used a water-filled circular tube (Saturn-Ring) worn around the head to perform dynamic neck exercise in addition to their normal football training. They increased their isometric strength and neck girth more than the control group of football training alone [14]. Neck strength training for the workplace [15,16,17] and military pilots [18,19,20] have also received some attention in the literature.

Since neck muscle strength training can reduce the risk of injury, applying the principle of traditional gravity trainer into the design of neck muscle training equipment was currently a way that could be seen (traditional weight trainer, TWT, Figure 1a). However, the physiological and anatomical structure of the neck were different from the limbs. The movement of range (MOR) of the neck was small with lack of a suitable grip or fixation function, thus it made this trainer a less pleasant experience to use. The complicated strength test system focusing on force or torque assessment, such as BIODEX and ISOMED, was operationally cumbersome and expensive, so it was not accessible to large populations in daily neck strength training (Figure 1b). In the sports and exercise practice, a simple training hat combined with steel wire rope connecting to weight objects (Hat Trainer, HATT, Figure 1c) was commonly used. Its operability, validity, and safety were relatively poor.

It is of great practical significance to develop new strength trainers for the safety and convenience of neck training. In fluid filed, the oscillatory hydraulic damper could produce a resistance opposite to the direction of rotation, and the resistance increases with an increase of the force, which almost will not generate inertia [21]. This resistance has the characteristics of softness and compliance, and may be more suitable for the strength training of the neck. Based on the characteristics of oscillatory hydraulic damping and anatomical structure of the human neck, we designed a new oscillating hydraulic trainer (OHT, Figure 2) for neck flexion and extension in the sagittal plane according to the morphological data of Chinese humans aged 18–65.

The purpose of this study was to evaluate the newly designed OHT of the neck through objective and subjective methods, and to compare it with the TWT and HATT. Objective measurement was to collect the surface electromyography (sEMG) of muscle activities. By comparing the activation degree and mode of relevant muscles in the neck exercise process, the influence of different trainers on the resistance traits of neck muscles was explored. Subjective ratings of the product usability were evaluated by referring to the satisfaction assessment model of Ren [22]. The proposed study is expected to provide more economical and safer training methods in national fitness and rehabilitation resistance training, and to prevent neck injuries for trainees.

### Design

Oscillating hydraulic trainer (OHT) is mainly divided into five parts: the oscillatory hydraulic damping device, adjustment mechanism, swing arm, electric height adjustment support post, and support base. This is shown in Figure 2. When the OHT is used, the exerciser’s neck joint and the rotating shaft of the oscillatory hydraulic damper are aligned in a straight line. The exerciser uses adjustment mechanisms to fix the head and chest baffles to facilitate neck exertion. When exercisers perform the neck flexion and extension exercise, the swing arm drives the rotating shaft of the damper, and the rotating blade rotates with it. The liquid in the corresponding cavity is compressed by the blade and then flows through the throttle into another cavity of the rotary cylinder. Since the liquid is pressed against the blade to create resistance in the opposite direction of the swing arm, a resistance training effect can be produced. The exercisers can adjust the throttle area through the throttle valve to change the damping resistance to fit their training needs.

The core component of the trainer is the oscillatory hydraulic damping device; as shown in Figure 2, the angular velocity output of the hydraulic damping can be described by the following equation:(1)$$\omega =\frac{c\alpha \sqrt{\frac{2}{\rho }\left(p-p_{0}\right)}}{R_{d}A}$$
where *ω* is the swing angular velocity, *c* is the throttle over current coefficient, *ρ* is the fluid density, *α* is the throttle area, *p* is the fluid pressure, *p*_0_ is the return pressure, *A* is the blade area, and *R_d_* is the oscillation cylinder equivalent radius [23]. From Equation (1), when the blade area and the liquid density are set to be constant, an increase in the force of the neck is manifested as an increase in the pressure of the pressed cavity. After the square root operation, the increase in the angular velocity is far less than the resistance increase. When the force of the neck decreased, the angular velocity decreased quickly. The damping device is safe for exercise because of almost no inertia [24]. The neck is one of the most flexible and unstable joints; it will be easy to be injured by a sudden external force [25]. Thus, it can be seen that no inertia is a quite advantageous factor in neck strength training.

According to the human morphological data of Chinese adults [26], we designed three adjustable modules in the OHT, which were electric height adjustment, neck fixation adjustment, and chest baffle adjustment mechanism. Before exercise, exercisers can adjust the modules to make the trainer fit the ergonomic positions. The adjustment range of electric height was 0–120 mm, the neck fixation adjustment distance was 140–212 mm, and the chest baffle adjustment range was 260–375 mm. In order to meet the need for neck flexion and extension training of different intensities, a 12-gear knob was designed to set the exercise resistance.

## 2. Materials and Methods

### 2.1. Subjects

A total of 12 healthy men were recruited as subjects to participate in the experiment. The basic information was as follows: age 23.8 ± 1.7 years, height 176.7 ± 5.2 cm, weight 69.3 ± 4.3 kg, no neck and shoulder musculoskeletal injuries. In order to ensure the validity of the experiment, subjects were required not to participate in any heavy physical activities within 24 h before the experiment to avoid muscle fatigue producing deviation in the experimental results. Before the experiment, all participants understood the purpose and process of the experiment, signed the informed consents, and registered their personal information.

### 2.2. Apparatus

#### 2.2.1. Neck Strength Training Trainers

##### The Simple Hat Trainer (HATT)

The simple hat trainer (HATT) could perform neck flexion and extension exercise with the steel wire rope, pulley, and barbell pieces. The resistance load was changed by adjusting the number of barbell pieces. It was purchased online.

##### The Traditional Weight Trainer (TWT)

The overall structure of the TWT was consistent with the aforementioned OHT. The training resistance was achieved by gravity pulling.

##### The Oscillating Hydraulic Trainer (OHT)

Subjects adjusted the resistance load by turning the knob on the hydraulic cylinder. The load range was from 1 to 12 gears, which could meet the needs of light, medium, and heavy resistance.

##### The Mobile Phone

A mobile phone (Mi10, M2001J2C, Xiaomi Corporation, Tianjin, China) was used to play the animation videos to guide the subjects’ neck flexion and extension speed.

#### 2.2.2. Data Acquisition Equipment

A multichannel physiological signal acquisition system (MP150, BIOPAC Inc, GC, Goleta, CA, USA) was used in the experiment to collect sEMG signal data for targeted muscles. Disposable Ag/AgCl electrodes were used, and the gel-based electrodes had a diameter of 30 mm and a distance of 2 cm between detecting electrodes. The sampling frequency of sEMG data acquisition was set to 1024 Hz.

### 2.3. Experimental Procedure

A 2-factor (3 trainers × 2 speeds) within subject experimental design was used in this study. Trainer types were the HATT, TWT, and OHT, and angular speeds of neck movement were 60°/s and 120°/s.

#### 2.3.1. Selection of Target Muscles

There were 4 muscles that could be used for sEMG detection in the neck joint, namely, semispinalis capitis, splenius capitis, sternocleidomastoid muscle (SCM), and upper trapezius (UT). However, according to the study of Queisser et al., sEMG signal detection of semispinalis capitis was greatly limited [27]. In order to determine suitable target muscles for sEMG signal collection, pre-experiments were performed prior to the formal experiment. It was found that the sEMG signal of the trapezius was much stronger than that of the splenius capitis during neck extension. Finally, the left/right sternocleidomastoid muscle (LSCM/RSCM) and the left/right upper trapezius (LUT/RUT) were selected as the target muscles.

#### 2.3.2. Maximal Isometric Voluntary Contraction and sEMG Collection

After skin was lightly abraded and swabbed with alcohol, the electrode was placed on the corresponding site of the target muscles. The sternocleidomastoid electrodes were placed adjacent to a point 30% of the distance from the sternal notch to the mastoid process and over the muscle belly of the sternal head. The upper trapezius electrodes were placed in the middle between the C6 spinous process and the lateral acromion [28]. Each pair of electrodes was spaced 2 cm apart. The placement of the electrode position is shown in Figure 3.

Subjects sat on the trainer and wore the HATT attached to a steel wire rope. The steel wire rope was connected ahead to the force sensor horizontally by a pulley, and the other end of the sensor was fixed to the ground. After the subject’s warming up moderately, the maximum isometric flexion and extension at the neck position of 0° were performed. The sEMG signals were recorded during experimental for the amplitude standardization of relevant muscles [29]. The MVC values of the neck extension were recorded for the determination of exercise load. The MVC for each action was measured 3 times, each time lasts 3 s, and the rest interval was 3 min.

#### 2.3.3. Selection of Training Load

Thirty percent of the average of the three times of MVC for each subject was used as the resistance load for the HATT and TWT [30]. Then, the subjects were asked to use the HATT to perform neck flexion and extension at a speed of 60°/s, guided by the speed animation video to experience the exercise intensity. Then, subjects adjusted the knob of the throttle valve of the OHT damper to coincide with the perception of similar load intensity under the same motion speed. The motion of range between flexion and extension was about −45°~45° in the sagittal plane, as shown in Figure 4.

#### 2.3.4. Experiment of Neck Flexion and Extension

After the end of all MVC tests, 20 min of rest was provided. Then, the subjects performed 5 neck flexion and extension training at 60°/s and 120°/s with 3 trainers (HATT, TWT, OHT). The speed was guided by the animation video. That is, each subject needed to complete 6 sets of neck flexion and extension exercise (3 trainers × 2 angular speeds), and the action angular range was −45~45°. The experimental process is shown in Figure 5. The sEMG signals were collected during 5 movements per set. In order to avoid generating fatigue, a rest of 10 min was provided after each set of exercise tasks being completed.

#### 2.3.5. Subjective Evaluation of Product Usability

After completing the 6 sets of exercises, subjects were also required to complete a subjective survey for the ratings of product usability referring to Ren’s research. The survey asked each subject to rate from the following three categories for product usability (A): (1) appearance usability (A_1_), including the aesthetically pleasing (A_11_), the sense of coordination (A_12_), and texture (A_13_); (2) performing usability (A_2_), including controllability of the product (A_21_), ease to adjust (A_22_), fault tolerance (A_23_), efficiency (A_24_), and man–machine (A_25_); (3) perceived usability (A_3_): product satisfaction (A_31_), reliability (A_32_), and comfortability (A_33_). Rating was on a 5-point scale: 1—very dissatisfied, 5—very satisfied. The indicators and weights are shown in Table 1.

### 2.4. Data Processing

#### 2.4.1. The sEMG Signals

In this study, the sEMG signal amplitude index root mean square (RMS) was used to represent the degree of muscle activation. The activation duration (D) and the peak timing (PT) of the sEMG signal waveform were used to characterize the muscle activation pattern.

##### Normalized Root Mean Square of sEMG (RMS%)

The RMS value of the sEMG signal was used to express muscle activation intensity [31]. In the presented experiment, each sample was cut into 10 segments of signal (5 flexion and 5 extension). The RMS value was calculated using the cut sEMG signal. The calculation formula was as follows:(2)$$RMS=\sqrt{\frac{1}{N}\overset{t+T}{\underset{t}{\sum }}EMG^{2}}$$
where *t* represents the sample start time of the signal data, and *t + T* is the sample end time. In order to reduce the differences between individuals, the sEMG amplitude index was usually normalized (or standardized) based on signals under MVC for analysis and comparison. The normalization process calculation formula was as follows:(3)$$RMS\%=\frac{RMS_{MEASURED}}{RMS_{MVC}}\times 100\%$$

In statistical analysis, the RMS% of left and right eponymous muscles were averaged to represent the neck flexor or extensor muscles.
(4)$${\mathrm{RMS}\%}_{\mathrm{Ext}}=\left(\frac{\sum_{1}^{5}\mathrm{RMS}\%\mathrm{LUT}}{5}+\frac{\sum_{1}^{5}\mathrm{RMS}\%\mathrm{RUT}}{5}\right)/2$$
(5)$${\mathrm{RMS}\%}_{Flex}=\left(\frac{\sum_{1}^{5}\mathrm{RMS}\%\mathrm{LSCM}}{5}+\frac{\sum_{1}^{5}\mathrm{RMS}\%\mathrm{RSCM}}{5}\right)/2$$

When analyzing and comparing the total amount of muscle activation in a single flexion and extension cycle with different trainers, the RMS% of neck flexor and extensor muscles were averaged to represent it.
(6)$$RMS{\%}_{Total}=\frac{RMS{\%}_{Flex}+RMS{\%}_{Ext}}{2}$$

##### Duration (D) and Peak Timing (PT)

The sEMG original signals were full-wave rectified, 4th-order Butterworth filtered, and 7 Hz cutoff. The length of muscle activity between onset and offset was defined as duration (D) (in Hull’s study, duration was expressed by bicycle crank angle). The onset and offset thresholds were determined as 20% of the peak amplitude of the sEMG signal. The detailed process referred to the research of Hull et al. [32]. The schematic diagram of each indicator is shown in Figure 6.

PT was expressed as a percentage with the following formula:(7)$$Peak\;Timing\%=\frac{Peak\;Timing-onset\;Timing}{Duration}\times 100\%$$

In the same manner as RMS%_Total_, D and PT were calculated by the average value of left and right muscles with five-time exertions. They were used to indicate the activation characteristics of the relevant muscles during neck flexion and extension.

#### 2.4.2. Subjective Evaluation of Product Usability

The subjective evaluations of the three trainers were carried out by product usability survey, which was calculated by taking the weights of apparent usability, performing usability, and perceived usability into account. The indicators and weights are shown in Table 1.
A_1_ = 0.267 × A_11_ + 0.332 × A_12_ + 0.401 × A_13_
A_2_ = 0.114 × A_21_ + 0.085 × A_22_ + 0.033 × A_23_ + 0.265 × A_24_ + 0.503 × A_25_
A_3_ = 0.176 × A_31_ + 0.385 × A_32_ + 0.439 × A_33_
A = 0.143 × A_1_ + 0.429 × A_2_ + 0.428 × A_3_

SPSS 26.0 was used for statistical analysis of all processed data.

## 3. Results

### 3.1. The sEMG Original Signals

Figure 7 shows the sEMG original signals and the rectified filtered waveform of a subject performing neck flexion and extension exercises with the three trainers. The original sEMG signal with the oscillating hydraulic trainer (OHT) demonstrated a remarkable difference that could be distinguished by the naked eyes from that of the simple hat trainer (HATT) and traditional weight trainer (TWT). The sEMG signals with the HATT and TWT were similar: The amplitude of the upper trapezius (UT) was quite obvious during the whole exercise process, and the signal was stronger when performing neck extension movement. The sternocleidomastoid muscle (SCM) had a small signal amplitude. Yet, the signal amplitudes with the OHT were obviously alternating. The sEMG signal waveform after rectification and filtering showed the difference more clearly among the three trainers. The UT muscles with the HATT and TWT displayed obvious contraction under neck flexion, but the SCM muscle signals were very weak. However, the UT and the SCM muscle with OHT were in a state of alternating exertion.

### 3.2. The Muscle Activation Intensity

As shown in Figure 8, when the subjects performed neck extension with the 3 trainers, whether it was at a speed of 60°/s or 120°/s, the RMS% of the agonist muscle upper trapezius (UT) was higher than that of the antagonist muscle sternocleidomastoid (SCM), *p* < 0.001. However, during the neck flexion, the RMS% of SCM muscle with the OHT was significantly higher than the HATT and TWT, *p* < 0.001. The results are shown in Table 2.

In order to observe the overall muscle activation when subjects performed neck flexion and extension with the three trainers at different speeds, the RMS% of the agonist muscles and antagonist muscles of neck flexion and extension were averaged for observation. The results are shown in Figure 9. The two-way repeated measurement ANOVA is displayed in Table 2. Because the data’s Mauchly’s Test of Sphericity was not assumed, we used Greenhouse–Geisser to make corrections. Statistical analysis showed that the speed effect on the RMS% was very significant during neck flexion and extension exercise. The RMS% during rapid flexion and extension was remarkably greater than that of slow flexion and extension (*p* < 0.001). The effect of three trainers on RMS% was also significant. The RMS% with the OHT was significantly greater than the other 2 trainers (*p* < 0.001). There was no significant difference of RMS% between HATT and TWT (*p* = 0.087, not displayed in the table). The interaction effect of speed and trainer on RMS% was very significant (*p* = 0.001).

### 3.3. The Muscle Activation Pattern

Hull used the waveform after sEMG signal rectification and filtering to reflect the activation state of leg muscles when the cyclist was paddling a bike. In this study, subjects were asked to perform neck flexion and extension exercise. Since the HATT and TWT only can give one-way gravity load ahead in this experiment, the resistance was only applied in the direction of neck extension during exercising. Therefore, the sEMG signal of the upper trapezius was processed during neck extension. Then, the duration (D) and peak timing (PT) were compared among the three trainers.

The D and PT of the sEMG waveform with the three trainers at two speeds are shown in Figure 10. The main effect analysis of the of D and PT on the exercise speeds and trainers are shown in Table 2. The D of agonist muscle differed significantly at different exercise speeds, but there was no significant difference among trainers. The interaction effect of speeds and trainers on D was very significant (*p* < 0.001). Further comparison of D among three trainers was used at different speeds. The D of sEMG waveform with the OHT was significantly shorter than that with the other 2 trainers at low speed (*p* = 0.002, not displayed in the table), but was longer than the other 2 trainers at high speed (*p* = 0.048, not displayed in the table).

There were significant speed and trainer effects (*p* = 0.004 and *p* < 0.001) on the PT during neck extension. The interaction of speed and trainer was not significant (*p* = 0.753). The PT of sEMG waveform was earlier at a faster neck extension than a slower extension in all trainers. Regardless of the speed of neck extension, the PT of sEMG waveform with the OHT was significantly later than that with the HATT and TWT.

### 3.4. Subjective Evaluations of Product Usability

The subjective evaluation results of the three trainers are shown in Figure 11. The statistical analysis showed that OHT had the highest product usability score (Table 3). There was a significant difference among the 3 trainers as a whole (*p* = 0.047). Apparent usability and perceived usability did not differ obviously among trainers (*p* = 0.215, *p* = 0.322). The performing usability of OHT was the highest among the 3 trainers (*p* = 0.014).

## 4. Discussion

There was research comparing the effectiveness of the pin-load neck exercise machine and elastic resistance bands; no significant difference was found [33]. In this study, the new designed neck oscillating hydraulic trainer (OHT) was compared with the two gravitational trainers HATT (simple hat trainer) and TWT (traditional weight trainer) from the perspective of resistance characteristics and product usability by combining objective measurement with subjective evaluation.

### 4.1. The Muscle Activation Intensity

Some of the roles and functions of the neck musculature may be not identical to those of the limbs or trunk, and thus the strength and conditioning needs of the neck might be different [34,35]. The HATT and TWT can only provide one-way resistance every time the exerciser performed neck flexion and extension. For example, in this experiment, the neck extensors were subjected to tension almost continuously, while the neck flexors bore no resistance, but were even subjected to tension of the direction of the flexion. Hence, the extensor muscles of the neck were continuously activated, while the flexor muscles were almost not activated. The OHT provided resistance in the opposite direction of the movement, thus it can provide two-way resistance load in one flexion and extension cycle. Figure 7 clearly showed the sEMG signal and filtered waveform characteristics of the three trainers.

RMS was an indicator that characterizes the magnitude of the amplitude of the sEMG signal, reflecting the degree of activation or exertion of local muscles [36,37]. It was normalized to reduce individual differences and facilitate comparison. The results of this experiment showed that when performing neck flexion and extension exercise, the HATT and TWT provided the neck extensor with gravitational resistance, so that the RMS% of the extensor, i.e., upper trapezius (UT), was much larger than that of the flexor, i.e., the sternocleidomastoid (SCM). Nevertheless, a different phenomenon was showed when subjects exercised with the OHT. The flexor SCMs were also significantly activated when the subject performed flexion and extension exercise with the OHT. That is to say, the UT and SCM muscles of subjects were simultaneously trained when performing exercise with the OHT. In particular, the overall activation degree of neck flexor and extensor muscles with the OHT (45.42%) was almost twice as much as that of gravitational one-way trainers HATT (25.62%) or TWT (26.14%) during rapid movement. Previous studies had pointed out that the size of the RMS of sEMG was determined by the combination of the recruitment of motor unit and muscle fiber action potential discharge during muscle contraction [38,39]. The level of muscle fiber action potential discharge increased, and more motor units recruited made muscles increase strength output in order to maintain a faster speed of movement. Therefore, in this experiment, when neck flexion and extension exercises were performed at a higher speed (120°/s), the activations of the muscles were more intense. The speed effect on muscle activation intensity was the greatest with OHT among the three trainers.

### 4.2. The Muscle Activation Pattern

The sEMG waveform after rectification and filtering could reflect the exertion characteristics of muscles. The duration (D) between sEMG onset and offset represented the length of time the muscle contracting. In this experiment, the HATT and TWT provided one-way gravitational resistance, thus the continuous contraction time of the neck extensor group was remarkably long during slow movement, and it was relatively shortened during rapid movement. Since the OHT provided passive hydraulic resistance, D of extension with OHT appeared shorter than that with the HATT and TWT when performed at a low speed. However, in the case of fast motion, gravitational trainers could provide greater resistance in the early stage, but might be a lack of sufficient resistance in the later stage due to mass inertia caused by high speed. The D of sEMG waveform in OHT under fast motion was significantly longer than that in the HATT and TWT. It showed that OHT can provide more adequate resistance stimulus for speed endurance training. This characteristic was especially suitable for strength training of athletes who overcome fluid resistance, such as swimming and aquatic sports. In these kinds of sports, athletes need to overcome the water resistance as consistently as possible in their range of motion [40], which was significantly different from the resistance characteristics of traditional gravitational trainers. This type of resistance gives muscles a longer activation time and is more adaptable to the training of the neck, which is of small movement and lacks the ability to grasp the handle of trainer.

Peak timing (PT) of muscle contraction has been investigated in previous studies. Wirianski A. described changes in EMG using duration, time from EMG onset to peak EMG, and peak time relative to the onset [41]. To some extent, PT reflected the structural characteristics of resistance, which received corresponding attention in specialized strength training. PT of muscle activation was more related to different competitive sports. In those sports requiring high explosive power, sporters needed to overcome the inertial resistance, so there exhibited shorter PT in rapid motion [42], while sporters in aquatic, swimming, road cycling fields and so on needed to overcome fluid resistance [43]. The compliance of fluid resistance was reflected in longer PT [44]. In this experiment, the PT of sEMG waveform in OHT was longer than that in the two gravitational trainers (HATT and TWT) under the condition of rapid neck extension. This resistance compliance characteristic shown by the long PT might have adequate advantages in terms of safety for fitness strength training in middle-aged and elderly people and rehabilitation training in injured patients. It is particularly meaningful when this kind of compliance resistance is used in muscle strength training for important human joints, such as the neck. Pilots would face an acceleration of high G-forces when driving a fighter jet [45]. It could be imagined that their neck strength training needs the assistance of trainers, such as the OHT.

### 4.3. Subjective Assessment of Satisfaction

Product usability was defined as “for users, the product can be effective, efficient and satisfactory to the level of specific goals” [46]. Though the user satisfaction evaluation method was subjective, its effectiveness was widely recognized [47]. The user satisfaction evaluation method plays a vital role in human factors research and design practice. In this study, subjects scored the highest product usability evaluation of the OHT (4.01 ± 0.35), which was significantly higher than that of the HATT (3.36 ± 0.91) and TWT (3.59 ± 0.46), *p* < 0.05. The results showed that users were satisfied with the OHT on the whole. The advantage of the OHT may be mainly reflected in performing usability (4.18 ± 0.47), which was higher than the HATT (3.24 ± 1.09) and TWT (3.58 ± 0.51), *p* < 0.05. The reason may be that the OHT only needs to rotate the damper’s throttle valve knob to achieve stepless adjustment of damping resistance. People will not give a high score for the evaluation of simple apparatus in accordance with human nature [48]. For the new neck trainer, the scores given by the subjects indicated that subjects recognized the product usability of the new designed trainer OHT in terms of psychology, appearance, and function.

## 5. Conclusions and Future Prospects

In this paper, a new neck oscillating hydraulic trainer was designed based on the oscillating hydraulic damper, and compared with the simple hat trainer and traditional gravitational trainer by objective test and subjective evaluation. Using the OHT can carry on two-way resistance training, thus the exercise volumes were almost twice as much as that using one-way gravitational trainers. The compliance of hydraulic resistance was safer and more suitable for neck strength training in multiple populations. The new neck OHT had high scores on performing usability and product usability evaluation, which reflected high user satisfaction and was more appreciated by users. Therefore, the OHT could provide a more efficient, safer, and better experience option for neck strength training.

The limitation with this study was that only healthy men were tested, while women and people with neck injury were not included in the experiment. Different groups using these trainers might have different sensations and effects; therefore the objective and subjective data in the experiment should be narrowly interpreted. The task carried out in the experiment was dynamic flexion and extension exercises in the sagittal plane, which was likely to limit the generalizability of the results to some extent. Therefore, in future studies, we would consider carrying out the test of various groups including women and patients with neck injury, and test of the neck lateral flexion (coronal plane) trainers with the same damping principle to verify the feasibility and validity of the new neck strength trainers.

## Figures and Tables

**Figure 1 healthcare-11-01518-f001:**
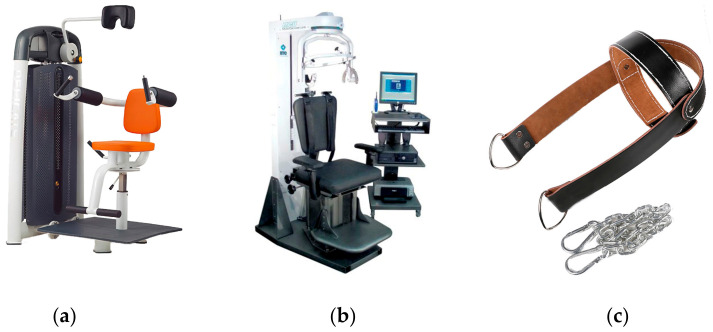
Three types of trainers commonly seen. (**a**) Traditional weight trainer; (**b**) Strength measurement system; (**c**) Hat trainer. The three pictures come from the Internet. The link of (**a**) was http://www.bj-orient.com/lists/content/id/348.html (accessed on 15 October 2022).The link of (**b**) was https://www.btetechnologies.com/products/functional-rehabilitation/multi-cervical-unit/ (accessed on 15 October 2022). The link of (**c**) was https://o2o.1688.com/offer/619823744763.html (accessed on 15 October 2022).

**Figure 2 healthcare-11-01518-f002:**
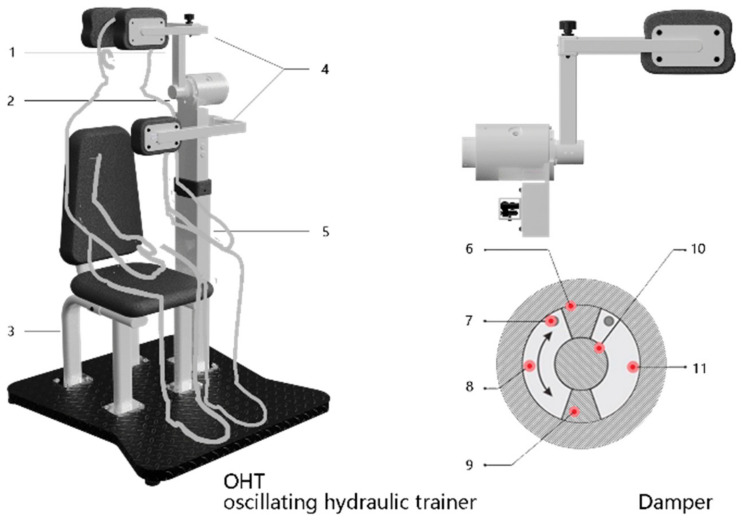
The illustrations of the OHT and functional structure of the damper. 1—Swing arm; 2—Oscillatory hydraulic damping device; 3—Support base; 4—Adjustment mechanism; 5—Electric height adjustment support post; 6—Fixed blade; 7—Throttle valve; 8—Cavity A; 9—Rotating blade; 10—Rotating axis; 11—Cavity B.

**Figure 3 healthcare-11-01518-f003:**
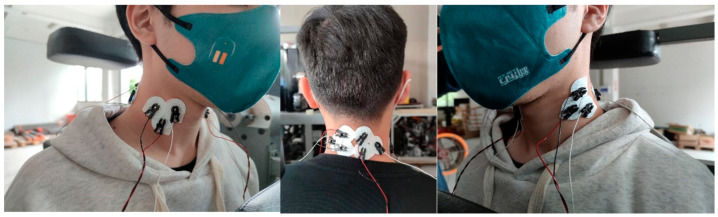
Target muscle and electrode placement schematic.

**Figure 4 healthcare-11-01518-f004:**
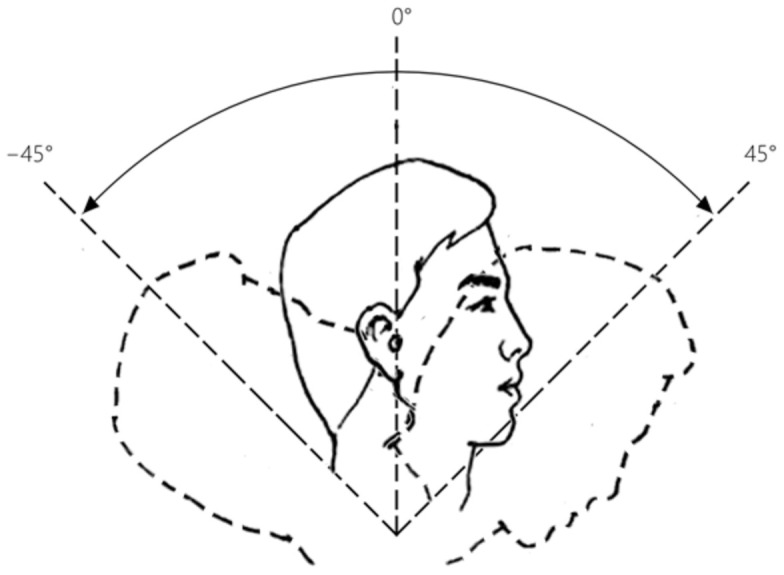
Range of neck flexion and extension.

**Figure 5 healthcare-11-01518-f005:**
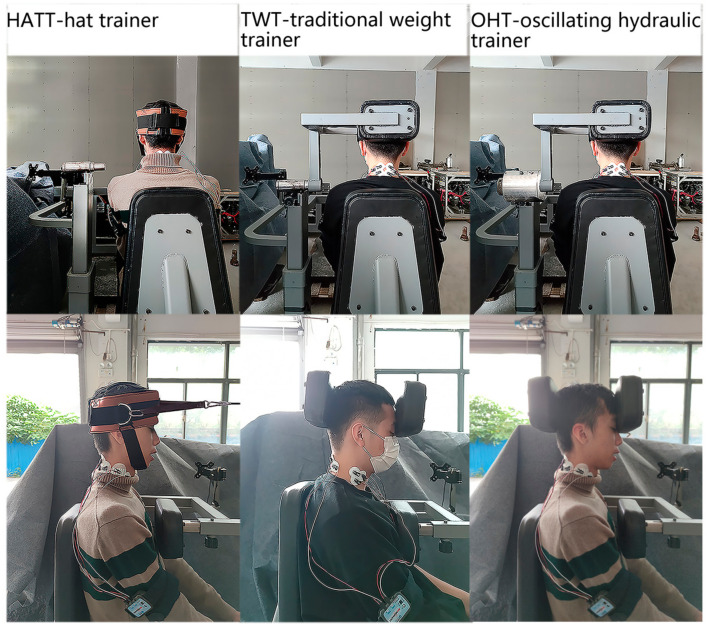
Illustration of the experimental process.

**Figure 6 healthcare-11-01518-f006:**
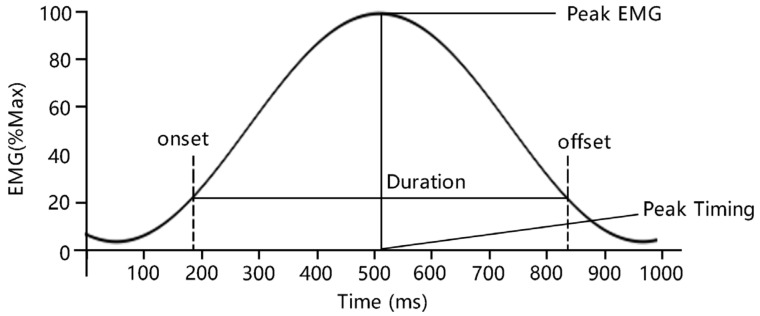
Illustration of muscle activation duration (D) and peak timing (PT).

**Figure 7 healthcare-11-01518-f007:**
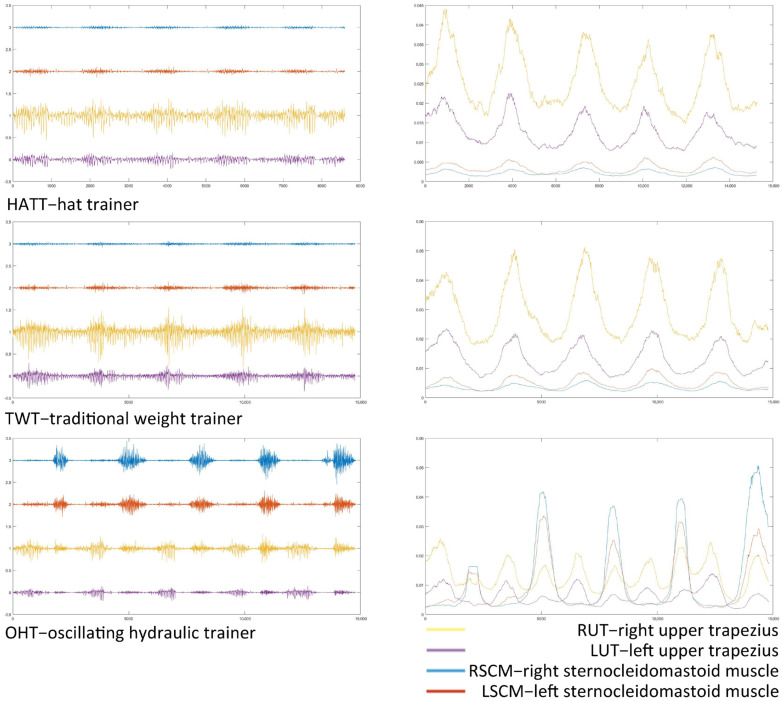
The sEMG original signals and rectified filtered waveforms.

**Figure 8 healthcare-11-01518-f008:**
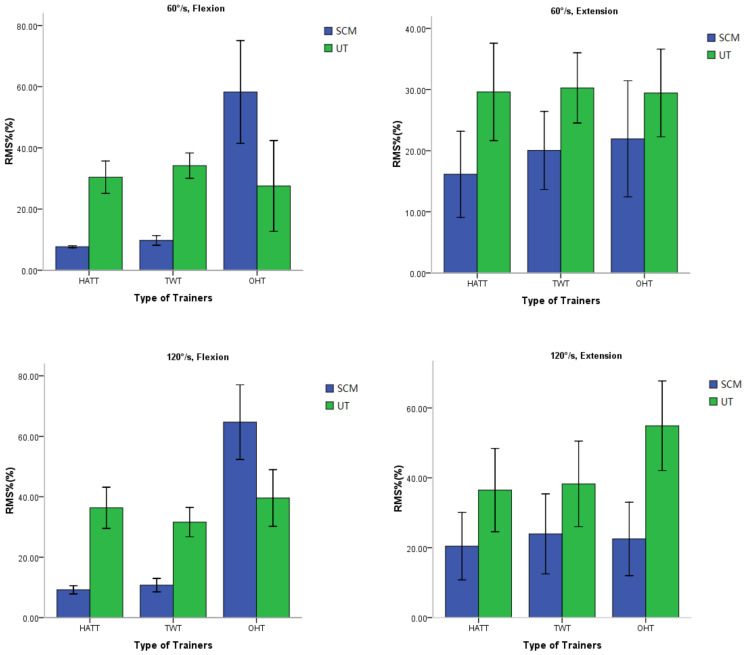
Flexion or extension RMS% of SCM and UT under three trainers at two angular speeds. The muscles were sternocleidomastoid muscle (SCM) and upper trapezius (UT). HATT: hat trainer; TWT: traditional weight trainer; OHT: oscillating hydraulic trainer.

**Figure 9 healthcare-11-01518-f009:**
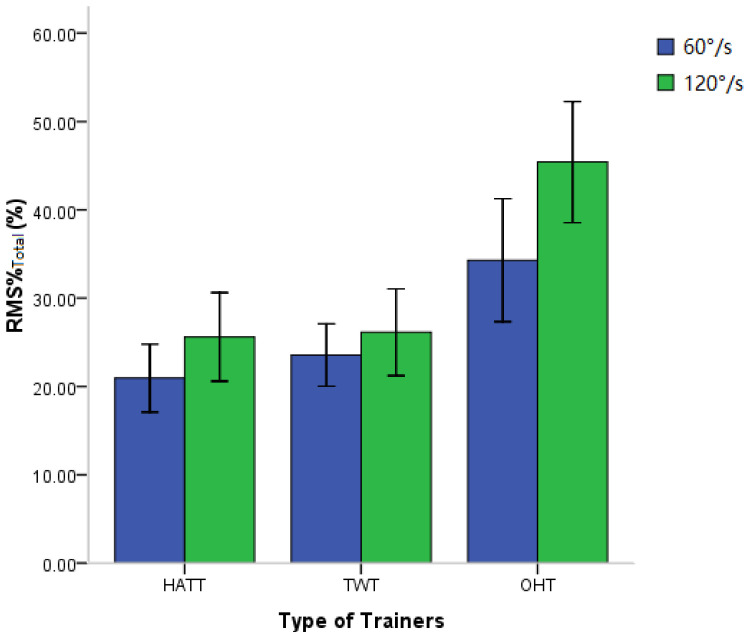
Total RMS% of SCM and UT under three trainers at two angular speeds. HATT: hat trainer; TWT: traditional weight trainer; OHT: oscillating hydraulic trainer.

**Figure 10 healthcare-11-01518-f010:**
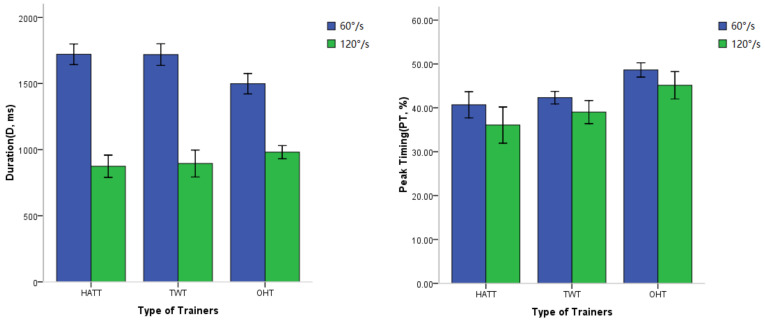
Duration and Peak Timing of upper trapezius during extension. HATT: hat trainer; TWT: traditional weight trainer; OHT: oscillating hydraulic trainer.

**Figure 11 healthcare-11-01518-f011:**
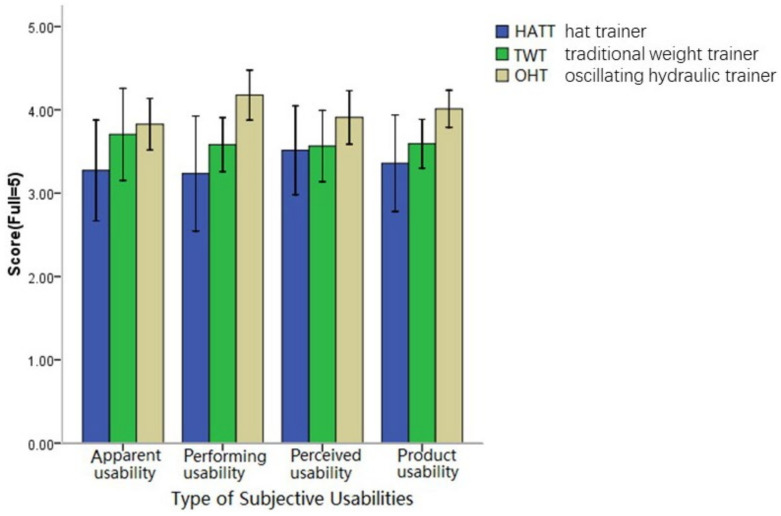
Subjective evaluations of three trainers.

**Table 1 healthcare-11-01518-t001:** Product usability indicators and weights.

Level-1	Level-2	Weight	Level-3	Weight
Product Usability A	Apparent Usability (A_1_)	0.143	Aesthetically pleasing (A_11_)	0.267
			Coordination (A_12_)	0.332
			Texture (A_13_)	0.401
	Performing Usability (A_2_)	0.429	Controllability (A_21_)	0.114
			Easy to adjust (A_22_)	0.085
			Fault tolerance (A_23_)	0.033
			Efficiency (A_24_)	0.265
			Man–machine (A_25_)	0.503
	Perceived Usability (A_3_)	0.428	Satisfaction (A_31_)	0.176
			Reliability (A_32_)	0.385
			Comfortability (A_33_)	0.439

**Table 2 healthcare-11-01518-t002:** Examination of the effects of angular speed and trainer type on sEMG parameters.

Parameter	60°/s			120°/s			Angular Speed		Trainer Type		Speed × Type	
	HATT	TWT	OHT	HATT	TWT	OHT	*p*-Value	η^2^	*p*-Value	η^2^	*p*-Value	η^2^
RMS% ^1^	21.24 ± 13.82	23.57 ± 12.10	34.30 ± 24.02	25.62 ± 17.31	26.14 ± 16.94	45.42 ± 23.64	0.000	0.371	0.000	0.285	0.001	0.173
D ^2^	1721 ± 123	1719 ± 128	1498 ± 121	875 ± 132	895 ± 161	981 ± 78	0.000	0.984	0.180	0.144	0.000	0.574
PT ^3^	40.68 ± 4.69	42.32 ± 2.25	48.64 ± 2.57	36.07 ± 6.48	39.04 ± 4.15	45.14 ± 4.92	0.004	0.535	0.000	0.681	0.753	0.026

^1^ RMS% was the average of the total flexions and extensions, %; ^2^ D was the activation time of the extension sEMG waveform, ms; ^3^ PT was the peak moment of extension sEMG waveform, %.

**Table 3 healthcare-11-01518-t003:** Examination of the effects of trainer type on subjective evaluations.

	HATT	TWT	OHT	F	*p*-Value
Apparent usability	3.27 ± 0.95	3.71 ± 0.87	3.83 ± 0.49	1.610	0.215
Performing usability	3.24 ± 1.09	3.58 ± 0.51	4.18 ± 0.47	4.902	0.014
Perceived usability	3.51 ± 0.84	3.57 ± 0.67	3.91 ± 0.51	1.175	0.322
Product usability	3.36 ± 0.91	3.59 ± 0.46	4.01 ± 0.35	3.370	0.047

## Data Availability

Data available on request due to restrictions eg privacy or ethical. The data presented in this study are available on request from the corresponding author. The data are not publicly available due to the reason that the university does not allow employees to publicize laboratory data without permission. The statistical results used by researchers to publish academic achievements are permitted by funding institutions.

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
