# Peer review of "An Evaluation Study of a New Designed Oscillating Hydraulic Trainer of Neck"

_healthcare, 2023, doi:10.3390/healthcare11101518_

Round 1

Reviewer 1 Report

A new type of damping neck flexion and extension training equipment was studied in this paper. In the context of the relative lack of research on neck strength training equipment, the study had certain research significance and practical value. The overall structure is clear and the language is unobstructed. The main problems are as follows:
1. For the design part of the equipment, it seems that there is no clear resistance corresponding to the 12-speed knob of the new neck trainer. So it is difficult for readers to form a more specific and clear concept;
2. Will the intensity perceived by the subjects be too subjective? And how to eliminate the differences between individuals?

Minor comments:
Keep the words consistent. For the indicators of product usability, in the previous paragraph, the values in A1, A11, A12 are the form of the lower corner. But in the subsequent paragraph, the values in A1, A11, A12 are not the form of the lower corner.

In conclusion, the article is recommended to be accepted.

Reviewer 2 Report

The study evaluates the effectiveness of a new oscillating hydraulic trainer (OHT) compared to two other trainers in terms of muscle activity and subjective measures to reduce the risk of neck injuries. The introduction provides sufficient background information on neck injuries, the importance of using a trainer, and the limitations of current trainers, and presents a clear description of the new OHT. The rationale and objectives of the study are well-defined.

However, I suggest improving the methods section by including information on the statistical analysis and the order of trainers. It would be helpful to know whether the order of trainers was randomized and how it may have affected the results. Additionally, the full spelling of abbreviations should be provided in the figure captions for clarity.

Overall, the results section provides a clear description of the study's outcomes. The results are properly interpreted and discussed in the discussion section. I find this study well-designed and the manuscript well-written, making a valuable contribution to the journal.

Reviewer 3 Report

An Evaluation Study of a New Designed Oscillating Hydraulic Trainer for Neck

The authors designed a novel device to strengthen the neck. The hydraulic device was tested on twelve subjects, and its efficacy was compared against the other two devices using electromyography measurements and subjective rating. The authors concluded that this novel device is suitable for neck strength training.

Main concerns:

The number of subjects evaluating the device is low. This situation negatively impacts the statistical power. Thus the statistical results could not be reliable.

Consent letters are missing because twelve human subjects were participating in a research study.

Minor concerns:

Several references should be added in Figure 1 to describe the source of the used images.

Figure 2 should include the anatomical position of the exerciser in the training device.
